# A Low-Cost Vision-Based Monitoring of Computer Numerical Control (CNC) Machine Tools for Small and Medium-Sized Enterprises (SMEs)

**DOI:** 10.3390/s19204506

**Published:** 2019-10-17

**Authors:** Hyungjung Kim, Woo-Kyun Jung, In-Gyu Choi, Sung-Hoon Ahn

**Affiliations:** 1Department of Mechanical and Aerospace Engineering, Seoul National University, Seoul 08826, Korea; hjkim81@snu.ac.kr (H.K.); afrak64@snu.ac.kr (W.-K.J.); timetraveler@snu.ac.kr (I.-G.C.); 2Institute of Advanced Machines and Design, Seoul National University, Seoul 08826, Korea

**Keywords:** machine tool monitoring, optical character recognition, open-source software, smart factory, small and medium-sized enterprises

## Abstract

In the new era of manufacturing with the Fourth Industrial Revolution, the smart factory is getting much attention as a solution for the factory of the future. Despite challenges in small and medium-sized enterprises (SMEs), such as short-term strategies and labor-intensive with limited resources, they have to improve productivity and stay competitive by adopting smart factory technologies. This study presents a novel monitoring approach for SMEs, KEM (keep an eye on your machine), and using a low-cost vision, such as a webcam and open-source technologies. Mainly, this idea focuses on collecting and processing operational data using cheaper and easy-to-use components. A prototype was tested with the typical 3-axis computer numerical control (CNC) milling machine. From the evaluation, availability of using a low-cost webcam and open-source technologies for monitoring of machine tools was confirmed. The results revealed that the proposed system is easy to integrate and can be conveniently applied to legacy machine tools on the shop floor without a significant change of equipment and cost barrier, which is less than $500 USD. These benefits could lead to a change of monitoring operations to reduce time in operation, energy consumption, and environmental impact for the sustainable production of SMEs.

## 1. Introduction

The Fourth Industrial Revolution, based on the latest advances in information and communication technology (ICT), is now leading the overall change in the way we live, work, and relate to one another [1]. Internet of Things (IoT) and mobile enable us to collect and transfer data, cloud and Big Data support store and analyze the collected data efficiently, and analytics, such as machine learning and deep learning, provide valuable information for decision making of human’s intellectual work [2]. In this new era of technological revolution, the smart factory is getting much attention as a solution for the factory of the future that represents an advanced manufacturing system able to integrate the entire production process including the way to design, produce, and market into ICT to deliver customized products at a minimum cost and time [3,4,5,6]. Also, this new factory model could play an essential role in reducing environmental impacts on sustainable manufacturing [7,8]. For example, Siemens’s Amberg factory has significantly reduced defect rates through IoT technologies and boasts the world’s best production quality rate at 99.99% [9]. GE’s Predix and Digital Twin support modern digital businesses by converging the industrial equipment, data, and the internet into a platform that enables the optimization of asset monitoring and management [2]. Adidas had opened a heavily automated manufacturing facility in Ansbach, Germany, called Speedfactory. This brand-new factory would pair a small human workforce with technologies, including 3-D printing, industrial robots, and computerized knitting to make running shoes [10].

Most manufacturing companies in developed countries have faced a lack of workforce and a decline in labor productivity, due to the aging population and low birth rates for the past decade. Thus, they are trying to improve productivity and stay competitive by adopting this smart factory concept [11,12]. Also, in leading manufacturing countries, various industrial policies for implementing smart factories are being executed to overcome this economic crisis, either by a government or private sector [4,13]. Yet, most small and medium-sized enterprises (SMEs), positioned at the third, or an even lower, layer of a supply chain in the manufacturing industry, still tend to be more labor-intensive and orientated towards short-term operations rather than long-term strategic issues since they do not have many resources [14,15,16]. Moreover, little research has been carried out on these challenges for SMEs participating in Industry 4.0 ecosystem [11,17]. Some researchers also pointed out that most SMEs have suffered the lack of expertise for IT and fear of the cost of failure to adopt emerging technologies [18,19]. In the Final Report of the Industry 4.0 working group, Kagermann et al. [20] reported that many SMEs are not prepared for the structural changes that Industry 4.0 will entail, either because they lack the requisite specialist staff or because of a cautious or even skeptical attitude towards a technology strategy that they are still unfamiliar with. In the management of the shop floors, forecasting return on investment of monitoring systems would be difficult in SMEs because of their short-term strategy from global competition and diversity of customer needs. Table 1 compares the major features between SME and large enterprises [18,21,22]. As a result, these challenges have made SMEs avoid adopting emerging technologies on their shop floor, although the smart factory and ICT are widely spreading.

### Monitoring of Machine Tool Status

The shop floor is the heart of manufacturing companies, and its operation determines the productivity and efficiency of their production process. The main challenges in effectively and sustainably running a shop floor are to ensure higher machine availability, reduce idle/downtime, and get a higher yield by continuously monitoring the shop floor environment. In general, conventional factories have limited visibility and communication between machines and operators on the shop floor, especially for SMEs. Because of the disconnection between most legacy machines and monitoring systems, operators on their shop floor are responsible for monitoring them. When an operator is running multiple machine tools on the shop floor, problems that cause shutdowns or stop machines cannot be identified until he or she arrives in front of those machines. Some safety codes to stop or pause the machine tools until the problem is corrected by operators also increase the unproductive idle time and energy consumption in the production sometimes. In this situation, the operator’s task is limited to monitoring only a few machines because she or he has to spend time watching these machine tools. These unproductive time for watching machines also increase the energy consumption of not monitored machine tools, as well as decrease the productivity of machining shop floor. Moreover, because most data collection systems are heavily reliant on paper forms in SMEs, this restricts the opportunity for timely decision making and data recording for future applications, such as historical data mining and optimization [23]. These challenges have led to the adoption of monitoring systems on the modern shop floors. Remote monitoring can reduce the unnecessary time and energy consumption for watching machines and allow time to be spent on other more valuable tasks, such as maintenance and optimization, on the shop floor.

Today, computer numerical control (CNC) machine tools, one of widely used processing equipment on the shop floor, transform the raw material into a complete part, which has free-from or sculptured surfaces, needed to make a product with a numerical control (NC) part program [24]. Conventionally, while setting up and running a machine tool on a machining shop floor, operators can monitor operational data via a specific human machine interface (HMI) of the machine tool and are responsible for checking an operational status and changing work pieces after machining a part [25]. Thus, to increase the utilization of machine tools, the operators have to visit their shop floor all the time [14]. In this manual and traditional approach, however, watching the machine tool status is hard to achieve because it sacrifices large number of labor and time [26]. Therefore, a real-time monitoring system needs to be considered incorporating shop-floor operations to reduce the idle time and the efforts for manual monitoring of machine tools which are unproductive on the part of employees or machines as a result of factors beyond their control [27]. The monitoring is also considered essential for reducing energy consumption and environmental impact, due to the increasing interest in the resource crisis and environmental pollution [28,29].

Monitoring of machine tools has been a subject of research interest for the last few decades [30,31,32]. Most common objectives of monitoring applications are tool condition monitoring and monitoring for maintenance. With additional sensors, such as laser beams, dynamometers, torque sensors, strain gauges, and vision cameras, conditions of components related to the accuracy and precision of the machining were identified, including tool wear and axes performance. Although these condition monitoring methods could identify conditions over the sensing ability of the operators and machine tools, they required expensive sensors, the expertise of machining knowledge, and increasing complexity of wirings. On the other hand, with the increasing use of modern machine tools on the shop floor, monitoring of machine tool status has been investigated [33]. In status monitoring methods, operating data retrieved from machine tool controllers, as well as additional sensors were collected and analyzed to identify an operational status of machine tools [30]. Data from control systems include motor currents, feed rate, and spindle speed, converted to valuable information with predefined rules or domain knowledge of engineers [34]. Also, commercial monitoring systems able to record the machine tool status and detect problematic issues have been launched by industrial vendors, such as Machine Tool Builders (MTBs) [35,36]. Recently, with increasing demands for sustainable growth, studies on monitoring of energy consumption were reported to reduce environmental impacts of operating the machine tools. Vijayaraghavan and Dornfeld [29] also reported a monitoring application to track maintenance states by analyzing the energy consumption of machine tools.

For digital image capturing in machine vision, Charge-Coupled Device (CCD) and Complementary Metal Oxide Semiconductor (CMOS) sensors have been developed for years. Recently, CMOS vision modules are widely used for consumer applications, such as mobile phones, tablets, and digital cameras. In the manufacturing industry, machine vision has become an essential tool to reliably identify a wide range of work pieces in the flow of goods within production processes. Recently, innovative machine learning and deep learning processes are boosting recognition rates and coverages. With falling costs and increasing average performance of vision devices, using a consumer-grade webcam is an apparent cost-saving and affordable approach in the machine vision system. For instance, various studies used webcams for motion detection, facial recognition, and fire-detection in the home automation systems [37,38]. For medical use, researchers have developed a non-contact monitoring system using a webcam to measure vital signs of patients as health parameters [39]. Also, webcams used to monitor how the 3D printer is printing an object [40].

In conventional approaches of using machine vision for the machine tools, the machine vision has been mainly used for monitoring tool condition and surface quality of machined part to achieve high productivity and precision of the machining. With advances in vision systems, surface roughness measured by correlating parameters, extracted from the vision system software processing [41], and small tool breakage hardly perceived by indirect in-process monitoring methods detected precisely [42]. These approaches were mainly focused on the quality of machining. On the other hand, monitoring systems for watching inside or outside of machine tools have been developed [43]. The users can monitor the operating status of machine tools through the online video via installed cameras. However, data on this remote watching only can be interpreted by human operators. In other words, the monitoring task was not reduced, but moved from the shop floor to remote places. Most operational statuses need to be captured from the machine tools through monitoring systems are almost same as information displayed on their HMI. Hence, if there is a system with a vision device that can adequately capture an image of the HMI and an open-source image processing can analyse the captured images, this system could provide an inexpensive monitoring service of machine tools.

Over the last decade, in addition, leading industrial vendors have tried to make their products scalable and cheap enough for SMEs [44]. In contrast, these monitoring systems are still designed for latest products and require specific software or expensive additional device have to be integrated with existing machine tools and effective implementation take a substantial amount of time; it can also be extremely rigid once installed [45]. At a machine tool level, Machine-to-Machine (M2M) communication protocols, such as MTConnect and OPC UA, have been considered as a potential standard. However, the use of these protocols is limited to certain classes of machines, and small Original Equipment Manufacturers (OEMs) do not have the capability to develop the software functions to integrate them into their product’s control system [46]. As a result, most of the existing manufacturing equipment, especially for SMEs, still do not have the built-in capability to transmit and receive data through a network connection [47]. By most estimates, the number of CNC machines capable of linking to the network is less than 10% of the installed base [48], and a lifetime of modern machine tools is known as more than 10 to 15 years [49]. Thus, these machine tools will be used, at least, a few more years with the current configurations because the cost of implementing communication protocols into legacy machines that could run a few tens of thousands of dollars is also a substantial barrier for SMEs.

Recently, thanks to advances in IoT technologies, new chances of real-time data collecting are opening [50,51]. In home automation and building management, various solutions with IoT have been proposed [52,53]. IoT devices were used for monitoring status, detecting problematic issues, and controlling their environments. Some researchers also reported monitoring systems for machine tools using IoT devices [26,54]. Moreover, the average cost of IoT devices is falling, and the recent trend towards free or open-source software allows for the development of a system that does not rely on proprietary and expensive commercial software [55,56]. At this point, Advanced ICT could be an alternative to reduce the entry barriers of adopting ICT systems on SME’ shop floors if they can be appropriately introduced and integrated on the shop floors. These benefits could also improve the responsiveness of SMEs to reduce energy consumption and environmental impact through digital transformation.

To monitor the operational status automatically in SMEs, experienced operators and expert knowledge are required, as well as additional sensors or communication devices and software, not equipped with most legacy machine tools. As mentioned earlier, these requirements have been barriers to adopt monitoring systems in SMEs. Despite all these challenges, to remain competitive and sustainable growth in the current market challenges of highly customized products and stiff global competition, manufacturing companies, especially SMEs, have to find a way to increase the productivity and reduce environmental impacts on their shop floor [57,58]. To boost the digital transformation in SMEs, the primary objective of this work is to develop an appropriate monitoring method without a significant change of the machine tools and expert knowledge to integrate them on the shop floor. Specifically, this study focuses on collecting and processing the operational data using cheaper and easy-to-use components, hardware and software, that are affordable for SME by using the latest advances of ICT. These goals lead us to consider a novel monitoring approach, KEM (keep an eye on your machine), using a low-cost vision device, such as a webcam and open-source platforms, small embedded computers and image processing libraries, which are lightweight, cost-effective, and flexible to configure a monitoring system.

Accordingly, the remainder of this paper is organized as follows. Section 2 introduces a vision-based data collection approach for machine tools and identification of the operational status. Section 3 describes the architecture of a vision-based monitoring system with software and hardware. In Section 4, a prototype of the proposed monitoring system is implemented, and the evaluation results of the prototype are described.

## 2. Vision-based Monitoring of Machine Tools

In this section, two main methods of the KEM, monitoring and identifying the operating status of machine tools, are described.

### 2.1. Monitoring Using a Webcam and Optical Character Recognition (OCR)

To collect the data from HMI of a machine tool, an optical character recognition (OCR) is one of the applicable technologies. The OCR is an algorithm that functions like a human’s ability reading [59]. The digital database of Google Books, for example, is made of converted data of the full text of books and magazine using the OCR. Figure 1 shows a basic workflow of the OCR. After getting a captured image from a vision source, image processing functions modify the image to be suitable for recognizing the character data. Then, the OCR algorithm identifies original expressions based on the trained data of recognizable characters.

In this work, Tesseract is selected as the OCR algorithm. Tesseract, maintained by Google, is one of most accurate free and open-source OCR engines currently available. To improve the result of Tesseract engine, grey scaling and binarization using an OpenCV (Open-source Computer Vision Library), one of best alternative open-source tools for the development of image processing and computer vision algorithms, are followed.

For implementing a prototype, Logitech C920r, which is highly rated and well-priced in consumer-grade webcams, was chosen. The price of the selected webcam is $99 USD, which is cheaper because the average webcam cost $20 to $300 USD in online malls. Table 2 shows the specifications of the selected webcam.

The selected webcam was tested to process the OCR of captured images of the HMI of a machine tool. In the result, the Logitech C920r showed good OCR results because of its higher resolution and capability of wide focus range. However, the proposed concept is not limited to specific brands or specifications of consumer-grade webcams. Any small camera or webcam can be used by following the user requirements. To figure out the feasibility of the low-cost vision-based monitoring, collecting operational data from for three different CNC HMI screens (FANUC, SIEMENS, and CSCAM) were evaluated, and results are shown in Figure 2. The results showed that the proposed approach is a universal approach which does not depend on a specific controller and can serve as a reliable monitoring service.

### 2.2. Identifying the Operating Status of Machine Tools

In the shop floor, the operators can identify the operational information of machine tools with the ability of vision and hearing. In actual field conditions, because of a noisy environment of machining shop floors, most operational data should be monitored through the HMI screen or network connections of monitoring systems.

In this research, a combination of the webcam and OCR replaces the reading ability of the operators. With a proper resolution of a webcam, operational data can be collected, including G&M codes, spindle speed, feed rate, and so on. Table 3 shows operational data and status, which can be collected from a general HMI screen of a machine tool. The availability of some parameters would depend on the functionality of the target machine tool.

In monitoring machine tools, the operational status means a higher-level condition of machine or operation which can be identified by collecting and analyzing the operational data [60]. Also, this information is essential for operating and managing machine tools, which could be useful when automated operations are required. Basic operational statuses, such as cycle start, and alarm can be acquired directly from one of the operational data, as shown in Table 3. These are changes (or events) at a point in time. On the other hand, identifying changes over a range of time or based on multiple conditions requires complex processing or time-series data with domain knowledge for reasoning, including spindle working (start and stop), cutting, and cycle time. Table 4 shows a list of reasoning logics and expected additional information regarding target operational statuses in this study. For example, the progress of operation can be represented by a value of NC code progress. However, the status of actual cutting can be identified by combining multiple values, such as actual feed rate and spindle speed. Also, monitoring of these operational statuses can help the users to know higher-level information, such as energy consumption or environmental impact in machining.

## 3. KEM Monitoring System

The KEM monitoring system is designed to improve the monitoring task by using the OCR with a webcam on the shop floor, as shown in Figure 3. Compare to the current workflow in SMEs, the proposed system can serve useful information to the users and ICT infrastructure, such as Big Data and machine learning, using advanced ICT and IoT devices.

The KEM monitoring system consists of two main modules: Client and server applications, as shown in Figure 4. The KEM client is responsible for collecting and uploading data, recognized from captured HMI images of a machine tool, and the KEM server stores the monitored data and process events of operational statuses of the machine tool. Each module is explained in the following sections.

### 3.1. KEM Client

With the region of interest (ROI) data, registered via ROI manager, the KEM client manage all data recognition and connection of a webcam by executing processes iteratively, as shown in Figure 5.

The KEM client has two sub-modules of ROI manager and OCR engine.

ROI manager: To analyze characters, ROIs of target items, based on the captured image, need to be defined, including locations (x and y), size (width and height), type (numeric or string), and a parameter for image processing. To support intuitive and easy-to-use of registering ROIs, a Graphical User Interface (GUI) with buttons and a list view of ROIs was designed. To specify the required input parameters, the user can capture an image from the video stream of the connected webcam and adjust a threshold value for pre-processing of the captured image. For enhancing the user experience, upper and lower boundaries of ROI also can be specified in this capturing process. A name of operational data is selected in a list menu, and the data type is set based on the predefined conditions, as shown in Table 2, automatically. Then, ROI manager converts subsets of the captured image by pre-processing, cropping, and image modifications, and transfers them to the OCR engine. All ROI information can be saved in a data file, and the file can be loaded for the same monitoring condition in future use.OCR engine: The OCR engine analyze subsets of a captured image using the OCR algorithm, the Google Tesseract. Results of the OCR process were converted based on the ROI data. Because of the imperfections of the OCR process, some abnormal values of the results are replaced as the previous one. This approach is a naive solution, but effective because most abnormal values would not be expected to change significantly within the sample period because of the dynamic properties of mechanical components, such as inertia and friction, in manufacturing equipment.

By integrating the ROI manager, OCR engine, and KEM client on a small computer with a connected webcam, the client configuration works as a smart sensor in the IoT network. To help the user understand the monitoring system, the graphical data of ROI and OCR results are displayed on a monitoring image in real-time. The upload frequency can vary from few seconds to a few minutes depending on the number of ROIs and the processing time of the OCR algorithm. To upload the monitored data to a KEM server, Wi-Fi and MQTT (Message Queuing Telemetry Transport) broker are used for the network communication. Server-side operations will do data processing and analysis.

### 3.2. KEM Server

In this section, the architecture of KEM server for processing real-time monitoring data from the KEM client is described. The architecture is composed of software modules, as shown in Figure 6.

Communication and data platform: To support standardized communication and data gathering from various smart sensors in IoT networks, Mobius platform is used as a data platform of the KEM server. Mobius is an IoT server platform complying with the oneM2M standards [61]. It provides all of the functionalities for IoT devices, including registration, data management and repository, device management, security, communication management and delivery handling, discovery, subscription, and notification. For interconnecting the Mobius and end-point devices, such as smart sensors, the Mobius provides bindings for MQTT using a wireless network. In the Mobius platform, Node.js and MySQL are used as a core development framework and Database Management System (DMBS) as a data storage, respectively.Data processing: In the monitoring of machine tools, the operational data can be treated as events that occurred either at a point in time or over a range of time. Using event processing technologies can support to detect particular patterns of higher-level abstract events, as well as simple events, and react to them in a real-time manner. With the increasing use of computing devices and network communications, techniques, such as Rule Engine (RE) and Complex Event Processing (CEP) have been investigated comprehensively over the last decades [62,63]. In this study, an automation module that includes RE and CEP is developed using the Python language for data analytics. These two sub-processes can be used to identify simple and complex events by the rule like IFTTT (IF This, Then That) or pattern matching using sliding time windows. For example, when an event occur, means the machine tool status is changed or goes over a specific limitation, the KEM server can react to send an alarm message to the users through SMS or email automatically. For managing events, Node-RED is used. Node-RED is one of the flow-based programming tools, developed by IBM, and widely used for integrating information flows of IoT devices [64].Data visualization: An online monitoring application via web pages is developed for data visualization. The advantages of web-based tools are not only familiar user interface, but also good accessibility to the resources and knowledge [65]. The users can access the monitored operating data through a web browser on their laptop or mobile phone. To display the data in a graphical form, such as chart and gauge, Node-RED Dashboard is used. Node-RED Dashboard is a front-end visualization tool for the Node-RED.

Regarding implementation, the KEM server is designed to run as either cloud computing or fog computing. With the approach of traditional cloud-centric computing, the KEM server can be located in the remote servers of Amazon, Google, Microsoft, as well as company-owned. However, the burden of uploading data to remote cloud datacenters is on the resource-constrained IoT devices, sometimes leading to inefficient uses of their bandwidth and energy. Malek et al. [66] pointed out that processing data in in-field devices could enable the scalability of the platform by reducing remote processing time, with low data transmission and storage. With these challenges of real-time data, IoT data processing activity is moving closer to IoT data sources or data sinks. Fog computing is a highly virtualized platform that provides computing, storage, and networking services between end devices (things) and traditional computing networks [67]. Thanks to the functionalities of Mobius that support a smart gateway approach and MQTT broker technologies, the KEM server can be executed on a small computing device, such as Raspberry Pi and LattePanda, as well as a personal computer on the local area (shop floor). Figure 7 shows a monitoring scenario of multiple shop floors based on the fog computing concept. Each local hub uploads the operational statuses to the remote cloud, then the cloud aggregates and processes the statuses of all connected shop floors.

## 4. Implementation and Evaluations

To evaluate the feasibility of the proposed monitoring system (KEM), a prototype testbed was implemented with a 3-axis milling machine, as shown in Figure 8.

By using a consumer-grade webcam, the service was able to monitor HMI of the machine tool cost-effectively. Although the webcam is not a standard device on the shop floor, it could be attached and removed with a simple structure on a machine tool because of its compact and lightweight form factor. The 3-axis milling machine is VESTA 660 10K, manufactured by Hwacheon, and is equipped with a controller model 0i–MD, manufactured by FANUC. The proposed system monitored the HMI screen when the operating mode was Manual Data Input (MDI) which is one of the modes the machine tools can be operated. The used part program was for machining small pockets on a steel plate, and the machining time was about 40 minutes.

All modules at the client and server sides were written in the Python language. Python is one of the general purposes and high-level programming languages. This language is an advanced scripting language that emphasizes code readability and enables developers to use fewer lines of code in comparison with Java or C++, especially in engineering and science fields. GUIs of KEM client were designed using Qt—one of the most popular cross-platform app development technologies around.

In the evaluation, the KEM client was executed on LattePanda as a microcomputer. The LattePanda contains Intel Atom Cherry Trail 1.8 GHz quad-core processor, contains 4 GB running memory, 64 GB built-in Solid-State Drive (SSD), and Windows 10 IoT Enterprise. Operational data were monitored, including line number of NC code, actual feed rate, and actual spindle speed. As a result, selected items in the HMI screen were successfully monitored via the webcam and the prototype software at 1 Hz, as shown in Figure 9.

The KEM server was configured using Raspberry Pi 3 Model B+ and Raspbian Operating System (OS), one of Linux OSs to run as a fog hub in the network. To identify the operational status, the automation module processes the monitored operational data using developed RE and CEP functions in the server side. Then, the identified statuses were stored in a database of the KEM server.

The time for configuring monitoring conditions took less than a half hour. Through the GUIs of the KEM client, all information was simply configured by clicking a mouse cursor on a webcam image and typing text or numerical values. All components were connected via the Internet using a Wi-Fi network. As a result, the user can access the monitored data and working status using a mobile phone or laptop, as shown in Figure 10.

The results revealed that the proposed monitoring system is easy to integrate and can be conveniently applied for legacy machine tools on the shop floor without a change of equipment, such as the integration of an additional electric circuit. Although the sampling rate was not as fast as industrial protocols, the monitored data were enough to show the operational status to the users at 1 Hz. Besides, the method of capturing the HMI screen via the webcam does not depend on any specific vision device and controller. Thus, the proposed concept has a potential of universal use for any numerically controlled equipment which has a proper display for human operators. This benefit could increase the availability of legacy machines for a particular task in a remote area without operator’s watching, saving cost to invest and resources, such as energy and materials, for building new machine tools.

In the monitoring task, real-time and automated monitoring from a remote place can significantly decrease the monitoring complexity of distributed machine tools on the shop floor. Hence, operators have the advantage of reduced monitoring task and can focus on more valuable tasks to complete the scheduled production plan. This change can also make a capacity to handle the environmental issues for the sustainable growth of SMEs. Moreover, using the event processing in the cloud, such as checking the status of the operation, can reduce the unproductive time until noticing the unwanted issues like alarms, as well as for watching machine tools.

Less integration of additional parts also reduced the unproductive time required for the installation. The installation process was sufficient to install prepared software turn on each device and configure basic settings similar to the installation of electronic appliances, which took less than an hour, compared to the conventional method took few days. This simple integration could not only improve the usability, but also reduces the fear of use for the field workers in SMEs. In addition, shorter installation time can reduce the downtime to be sacrificed for the installation in the production.

From an economic point of view, the direct costs of the proposed system highly depended on the hardware used. Most of the software and libraries used in the implementation of the prototype were freeware or open-source software, but showed enough performance comparable to commercial products in this study. Hence, the Total Cost of Ownership (TCO) in the evaluation was less than $500 USD, including power adapters and memory cards, compared to the conventional method, required several thousand dollars. Detailed information regarding the direct costs used in the implementation is listed in Table 5.

## 5. Conclusions and Future Works

A monitoring system of machine tools on the shop floor using a low-cost webcam and open-source software, such as Python for the application development, Google Tesseract for the OCR process, OpenCV for image processing was proposed. To evaluate the feasibility of the proposed monitoring system, a prototype was implemented and evaluated with the typical 3-axis CNC milling machine.

The evaluation confirmed that HMI of machine tools could be monitored via a low-cost webcam, which leads to easy integration and reduction in time and difficulty, needed for changing control logic and wire configurations of the control system. The results revealed that the proposed monitoring system is easy to integrate and can be conveniently applied to legacy machine tools on the shop floor without a significant change of equipment and cost barrier. Besides, the proposed concept showed a potential for universal use for any numerically controlled equipment which has a proper display for human operators. This benefit also leads to easy integration and reduction in time and difficulty for installation and use. The time required for installation was reduced by not the integration of additional parts, which took less than an hour, and the Total Cost of Ownership (TCO) was less than $500 in the evaluation, compared to the conventional method required several thousand dollars. Moreover, it was also identified that using open-source platforms which are lightweight, cost-effective, and flexible to configure can boost the speed of software development in few weeks and reduce the barrier to entry to the digital transformation for SMEs. These benefits could lead to a change of monitoring operations to reduce time in operation, energy consumption, and environmental impact for the sustainable production of manufacturing companies.

Following this study, more types of manufacturing equipment, such as turning centers, parts feeders, and 3D printers will be investigated for the potential use of the proposed approach. To meet the digital transformation with the industry 4.0 and smart factory, adapting advanced ICT, including Big Data analysis, machine learning, and CPS, is required.

## Figures and Tables

**Figure 1 sensors-19-04506-f001:**
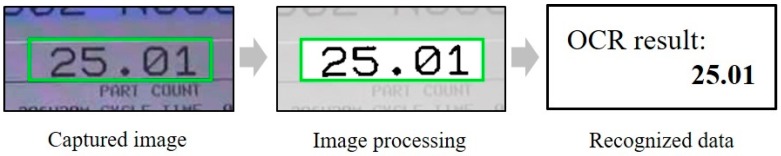
The workflow of optical character recognition (OCR) process.

**Figure 2 sensors-19-04506-f002:**
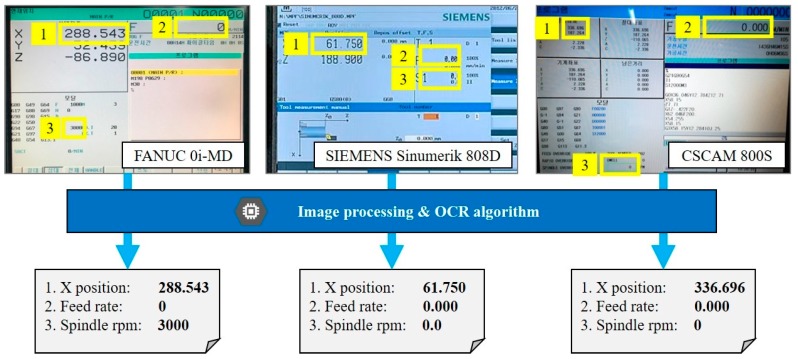
Example of collected operational data from three different human machine interface (HMI) screens.

**Figure 3 sensors-19-04506-f003:**
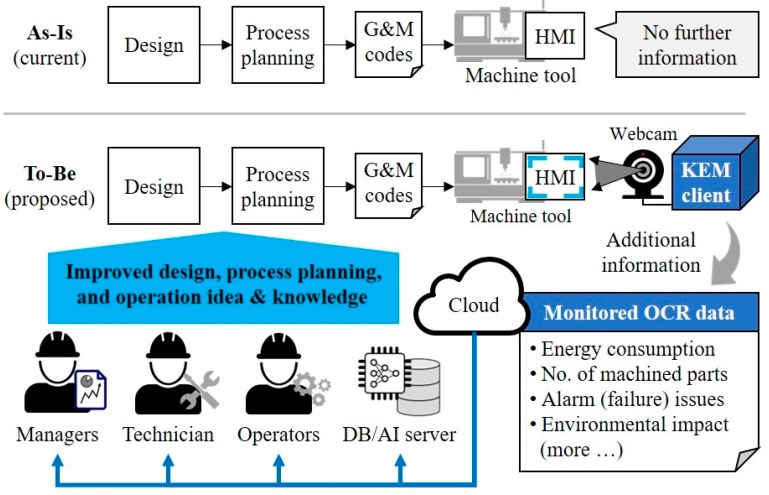
Comparison of the current status and proposed monitoring system.

**Figure 4 sensors-19-04506-f004:**
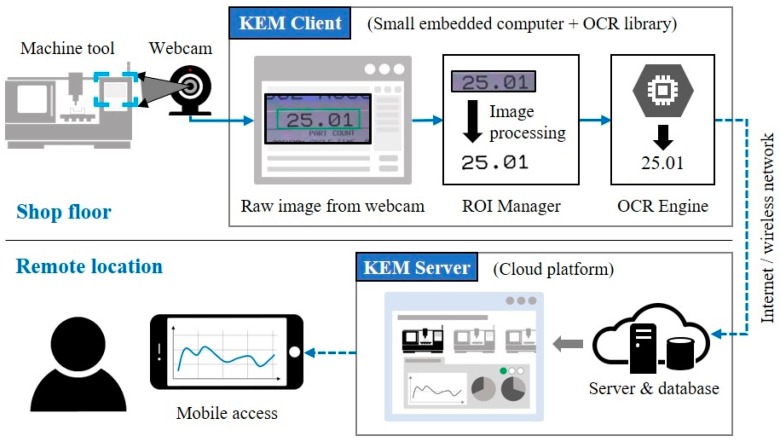
Configuration of the proposed KEM (keep an eye on your machine) monitoring system.

**Figure 5 sensors-19-04506-f005:**
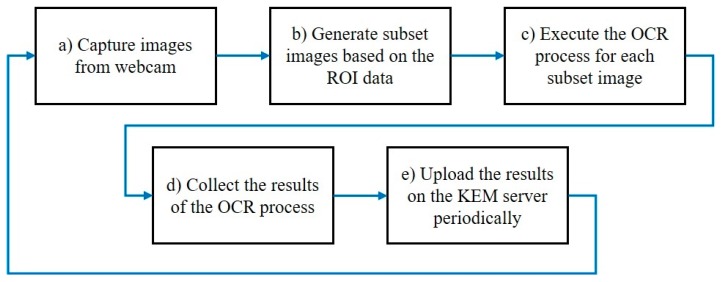
Working procedure of the KEM client.

**Figure 6 sensors-19-04506-f006:**
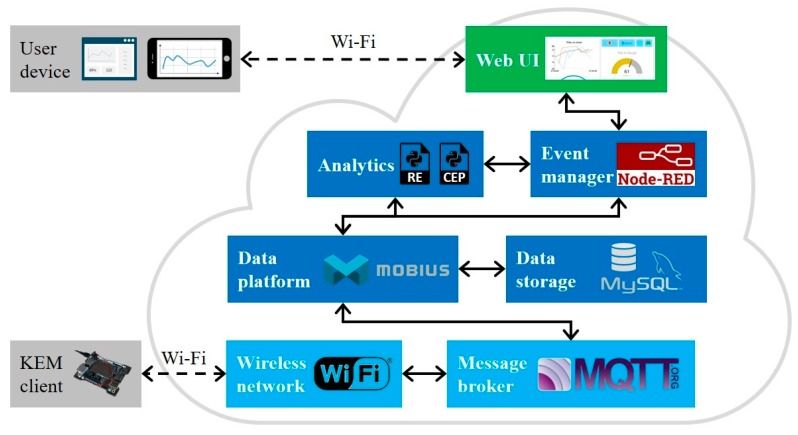
The architecture of the KEM server.

**Figure 7 sensors-19-04506-f007:**
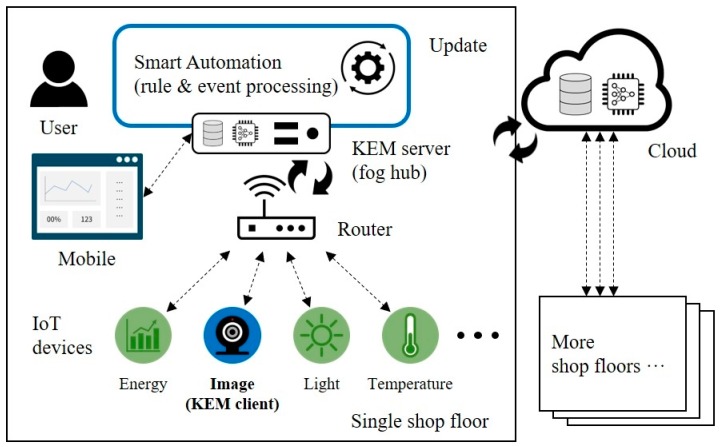
Monitoring scenario of multiple shop floors using the proposed system.

**Figure 8 sensors-19-04506-f008:**
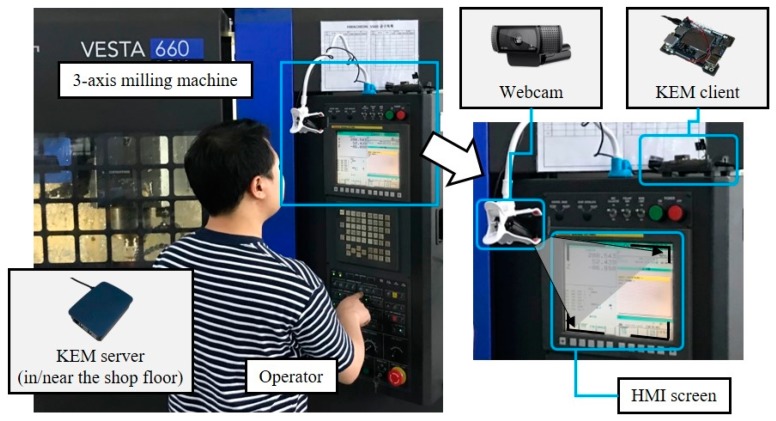
Configuration of the prototype evaluation.

**Figure 9 sensors-19-04506-f009:**
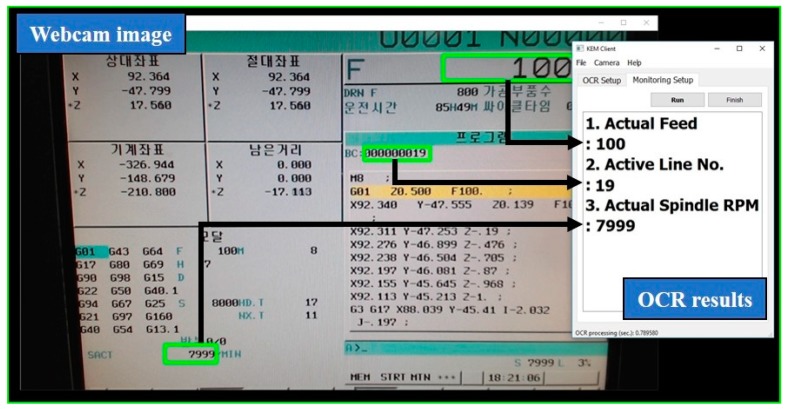
Display of OCR process in the KEM client; green rectangular boundaries mean the ROIs.

**Figure 10 sensors-19-04506-f010:**
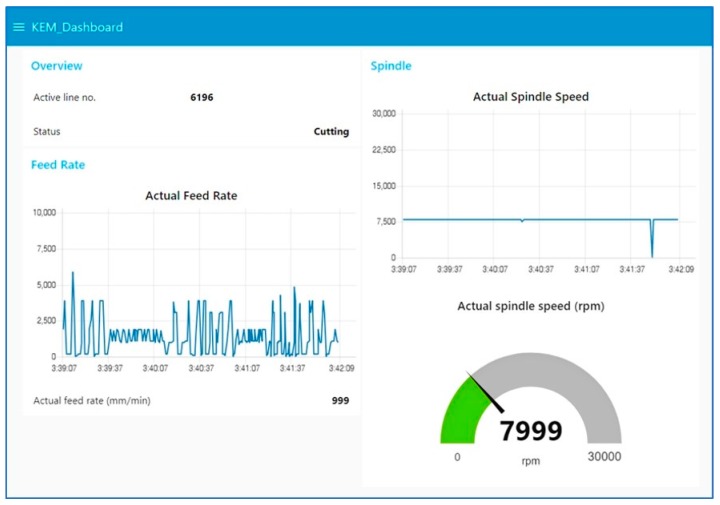
Online dashboard of monitoring from the KEM server.

**Table 1 sensors-19-04506-t001:** Comparison of features between small and medium-sized enterprises (SME) and large enterprise.

Topic	SME	Large Enterprise
Employees	Less than 250	250 or more
Business strategy	Market niches	Large market share
Production	Simple and flexible, labor-intensive with limited resources	Complex and rigid, capital intensive
R&D	Short-term and intuitive, lack of expertise, especially IT staff	Long-term and planned, a high number of researchers and experts
Procurement	Highly depends on external orders	Mostly independent from external orders

**Table 2 sensors-19-04506-t002:** Specifications of the selected webcam.

Model	C920r
Manufacturer	Logitech
Sensor type	CMOS
Resolution (pixels)	1920 × 1080
Frame rate (frames per second, fps)	30

**Table 3 sensors-19-04506-t003:** Operational data and status which can be collected from the HMI screen.

Information in HMI Screen	Acquirable Operation Data	Acquirable Operation Status
G&M Codes	Current line number, modal (m code) value	Program progress, cycle start and finish, takt time and elapsed time in machining, coolant use (on/off)
Spindle Speed	Values of program, actual, override	Spindle start and stop, in-cutting, machine idle
Feed Rate	Values of program, actual, override	Machine idle
Cutting Tool	Active tool number	(n/a)
Spindle load (optional)	Cutting load value	In-cutting
Alarm code (optional)	Error messages or codes	Reason of alarm (failure) in machining

**Table 4 sensors-19-04506-t004:** Reasoning logics and expected additional information on the operational status.

Operational Status	Reasoning Logic	Expected Additional Information
Cycle start	Line number changes from 0 to 1 or higher	Time of cycle start
Spindle Start	Actual spindle speed changes from 0 to higher	Working in cutting status
Cutting	Actual feed rate > 0 and spindle speed > 0	Machine-in-use
Spindle Stop	Actual spindle speed change from any to 0	Working in non-cutting status
Cycle finish	M30 or M02	Time of cycle finish and no. of machined parts
Machine idle	Spindle speed is 0, feed rate is 0, and keep these conditions more than 5 s	Reducing energy consumption
Alarm	Refer a list of alarm code	Maintenance issue
Takt time	Cycle finish time—cycle start time	Productivity
Elapsed time	Current time—cycle start time	Energy consumption
Coolant use	Time of coolant on	Monitoring and reducing environmental impact

**Table 5 sensors-19-04506-t005:** Direct costs of the prototype implementation.

Item/Tool	Product/Service	Cost (USD)
Webcam	Logitech C920r ^1^	$99
Client computer	LattePanda (4G/64G, Windows 10 IoT Enterprise) ^1^	$209
Hub computer	Raspberry Pi 3 Model B+ ^1^	$35
Python IDE	Microsoft Visual Studio Code ^2^ and Python IDLE (v3.6.4) ^3^	$0
GUI design	Qt Creator and PyQt ^3^	$0
OCR	Tesseract (v3.5.1) ^3^ and tesserocr (v2.2.2, python wrapper package) ^3^	$0
Image processing	OpenCV ^3^	$0
Data platform	Mobius IoT platform (v2.0) ^3^	$0
Web chart	Node-RED Dashboard ^3^	$0
Communication protocol	MQTT ^3^ and onoM2M ^3^	$0
Wireless network	Wi-Fi (hardware supported, Raspberry Pi and LattePanda)	$0
Cable, etc.	Power connector, holding device, and so on	$50
	(Total sum)	$393

Note: ^1^ commercial, ^2^ shareware, and ^3^ open-source.

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
