# Peer review of "A Low-Cost Vision-Based Monitoring of Computer Numerical Control (CNC) Machine Tools for Small and Medium-Sized Enterprises (SMEs)"

_sensors, 2019, doi:10.3390/s19204506_

Round 1

Reviewer 1 Report

This paper proposes a low-cost monitoring system for CNC of small and medium-sized enterprises vividly, which is easy to integrate and has the application foreground due to the cost advantage. Some observations with regard to the paper are given below: 1.There are some expressions that seem not to be accurate, such as “attend their shop floor” in Line 75, “Materials The shop floor” in Line 105, “…with sensing ability of…” in Line 239, and the inconsistent tense of the sentence in Line 385. It is recommended to modify the writing and improve the quality. 2.As for the contents of Section 2 and the part of Section 3 before section 3.1, I think that these analysis and discussion about status monitoring and machine vision belong to the Introduction part. They are nothing new developed but some existing references/technologies summation. It is suggested these contents should be rewritten in Section 1, so that the Introduction could be more comprehensive. In addition, is the contribution of this proposed approach a novel integration structure of existing intelligent methods? I am confused whether the detailed methods in Section 3 introduced are developed by the authors or by other scholars. Section 4 also makes me feel the same. 3. In the sentence starting at Line 250, what does “a higher-level condition of machine or operation” mean specifically? The statement is not clear enough to understand this advantage of operational status. 4.The References list should be modified,"Mission reliability evaluation based on operational quality data for multistate manufacturing systems","Risk-oriented assembly quality analysing approach considering product reliability degradation" and other related published papers in Sensors like "Risk-Oriented Product Assembly System Health Modeling and Predictive Maintenance Strategy","A Mission Reliability-Driven Manufacturing System Health State Evaluation Method Based on Fusion of Operational Data" and etc. should be cited.And the conference papers in the references list should be deleted substantially.

Author Response

We sincerely appreciate your kind and detailed review and comments on several important points in this paper. 

We provide our response as an attachment.

Reviewer 2 Report

The paper describes the use of an imaged based sensor with optical character recognition to monitor the status of a CNC machine.

The topic is highly relevant and emerging and the work is clearly situated with the state-of-the-art.

Author Response

We sincerely appreciate your kind and detailed review and comments on this paper.

We modified the article according to the opinions of other reviewers.

Thank you again for your kind review.

Reviewer 3 Report

Comments and Suggestions for Authors
(will be shown to authors)

Authors present an interesting application for monitoring a machine tool with a digital HMI. They use know techniques and commercial devices to implement a monitoring device that can be useful to a limited segment of machines: those that have a screen where the data of the process is displayed, but whose HMI software is not prepared to communicate with external equipment. Here there are two main points to clarify:

The estimation of the number of machines capable of linking to the network (10%, line 155) was made eighteen years ago. Although the lifetime of a machine it is said to be known as more than 10 to 15 years, looks like most of those machines should be now out of use. Further, although the machines were not prepared to connect to a network, they could be prepared to connect to a single device and allow the communication of the data to be monitored.

With all, an estimation of the answer to the question: “How many of the controls for machine tool sold in the last ten years do not allow the communication between the HMI and an external device?” will help to see the range of application of the presented device.

Author Response

(The authors gave the same response as above.)

Round 2

Reviewer 3 Report

Authors present an interesting application for monitoring a machine tool with a digital HMI. They use know techniques and commercial devices to implement a monitoring device that can be useful to a limited segment of machines: those that have a screen where the data of the process is displayed, but whose HMI software is not prepared to communicate with external equipment.